# The conserved σ<sup>D</sup> envelope stress response monitors multiple aspects of envelope integrity in corynebacteria

Elizabeth M. Hart[1,2], Evan Lyerly[1,2], Thomas G. Bernhardt[1,2]*

1 Department of Microbiology, Harvard Medical School, Boston, Massachusetts, United States of America,
2 Howard Hughes Medical Institute, Harvard Medical School, Boston, Massachusetts, United States of America

* thomas_bernhardt@hms.harvard.edu

**Data Availability Statement:** The authors confirm that all data underlying the findings are fully available without restriction. All relevant data are within the paper and its Supporting Information

## Abstract

The cell envelope fortifies bacterial cells against antibiotics and other insults. Species in the Mycobacteriales order have a complex envelope that includes an outer layer of mycolic acids called the mycomembrane (MM) and a cell wall composed of peptidoglycan and arabinogalactan. This envelope architecture is unique among bacteria and contributes significantly to the virulence of pathogenic Mycobacteriales like *Mycobacterium tuberculosis*. Characterization of pathways that govern envelope biogenesis in these organisms is therefore critical in understanding their biology and for identifying new antibiotic targets. To better understand MM biogenesis, we developed a cell sorting-based screen for mutants defective in the surface exposure of a porin normally embedded in the MM of the model organism *Corynebacterium glutamicum*. The results revealed a requirement for the conserved σ<sup>D</sup> envelope stress response in porin export and identified MarP as the site-1 protease, respectively, that activate the response by cleaving the membrane-embedded anti-sigma factor. A reporter system revealed that the σ<sup>D</sup> pathway responds to defects in mycolic acid and arabinogalactan biosynthesis, suggesting that the stress response has the unusual property of being induced by activating signals that arise from defects in the assembly of two distinct envelope layers. Our results thus provide new insights into how *C. glutamicum* and related bacteria monitor envelope integrity and suggest a potential role for members of the σ<sup>D</sup> regulon in protein export to the MM.

## Author summary

Bacteria within the Mycobacteriales order, which includes the pathogen *Mycobacterium tuberculosis*, have a unique multilayered cell surface architecture. How they sense and respond to defects in the construction of this distinct envelope to maintain homeostasis remains poorly understood. Here, we used the model organism *Corynebacterium glutamicum* to reveal that the conserved σ<sup>D</sup> envelope stress response pathway responds to two distinct signals resulting from defects in different envelope layers. The results provide new insight into how members of the Mycobacteriales monitor the integrity of their surface

files. All NGS raw data from this study are available under BioProject PRJNA1113594.

**Funding:** This work was supported by Howard Hughes Medical Institute investigator funds (to T. G.B) and the Helen Hay Whitney Foundation postdoctoral fellowship (to E.M.H). E.M.H received salary from HHMI and HHWF and E.L received salary from HHMI. T.G.B. received salary from HHMI. The funders had no role in the study design, data collection and analysis, decision to publish, or preparation of the manuscript.

**Competing interests:** The authors have declared that no competing interests exist.

and pave the way for the identification of novel vulnerabilities in envelope biogenesis that may be useful for targeting with antibiotics.

## Introduction

The Mycobacteriales order of bacteria includes virulent microbes like *Mycobacterium tuberculosis* (*Mtb*) and *Corynebacterium diphtheriae*, opportunistic pathogens like *Mycobacterium avium* and *Mycobacterium abscessus*, and environmental species like *Corynebacterium glutamicum* (*Cglu*) and *Mycobacterium smegmatis (Msmeg)*. These Gram-positive, acid-fast bacteria share a unique diderm cell envelope architecture that distinguishes them from other microbes. Surrounding the inner membrane is a cell wall composed of peptidoglycan (PG) with covalently attached polymers of arabinogalactan (AG). Enveloping the thick cell wall is an outer membrane structure called the mycomembrane (MM), which is principally composed of long chain fatty acids called mycolic acids. The inner leaflet of the MM is composed of free mycolic acid as well as mycolic acids esterified to arabinan residues of the AG component of the cell wall. The outer leaflet is made up primarily of trehalose monomycolate (TMCM), trehalose dicorynomycolate (TDCM), and free mycolic acids [1,2]. In the model organism used in this study, *C. glutamicum*, the MM and the arabinan layer of the AG are non-essential, allowing us to genetically disrupt these synthesis pathways. The envelope plays essential roles in the growth and antibiotic resistance of many pathogenic members of the Mycobacteriales order [3,4]. Understanding the molecular details of envelope assembly in these organisms therefore promises to reveal new vulnerabilities in the process to target for antibiotic development.

Despite the crucial role of the coryne- and mycobacterial cell envelope for viability, pathogenicity, and antibiotic resistance, many aspects of its assembly remain poorly understood. Of particular interest are the pathways that regulate the formation and maintenance of the MM, the biogenesis of which is targeted by front-line anti-mycobacterial therapies [1,5]. The MM has embedded proteins called mycolate outer membrane proteins (MOMPs) that are thought to function in membrane transport analogous to the β-barrel porins in the outer membrane of Gram-negative bacteria [6]. However, few mycolate outer membrane proteins (MOMPs) have been functionally characterized. In *Cglu* and *Msmeg*, MOMPs are post-translationally modified with a mycolic acid molecule in a process termed *O*-mycoloylation that is required for assembly and/or retention of MOMPs in the MM [7–10]. This modification is performed by mycoloyltransferases, which transfer mycolic acid to different acceptor molecules (TMCM, AG, or proteins) [11, 12]. In *Cglu*, Cmt1 (Cgp_0413) is responsible for protein *O*-mycoloylation [7,13] whereas the identity of the mycoloyltransferase(s) responsible for protein *O*-mycoloylation in *Msmeg* remains to be identified [10]. It is hypothesized that MOMPs are also *O*-mycoloylated in *Mtb*, however this has not been formally demonstrated [10]. Importantly, the system that assembles β-barrel outer membrane proteins in Gram-negative bacteria is not conserved in the Mycobacteriales and the pathways that mediate MOMP transport and assembly into the MM remain unknown.

To identify proteins involved in MOMP assembly into the MM, we developed a fluorescence-activated cell sorting (FACS)-based screen for mutants defective in the surface exposure of a MOMP called PorH in *Cglu*. Insertions in the alternative sigma factor *sigD* and components of its activation pathway were among the strongest hits in the screen. These findings indicated that members of the σ<sup>D</sup> regulon may encode key factors in MOMP transport and/or MM integration. Although the screen did not identify the specific MOMP assembly factors we were seeking, the results helped us begin a further characterization of the σ<sup>D</sup> pathway, which

identified the missing site-1 protease that initiates cleavage of the anti-sigma factor RsdA to trigger transcriptional activation by σ$^D$. Using genetic perturbations and chemical treatments, we demonstrate that the σ$^D$ pathway is activated by what appear to be distinct AG and mycolic acid biosynthetic defects, providing insight into the envelope precursors that the signaling system is likely monitoring to regulate its activity. Given the conservation of the σ$^D$ pathway among the Mycobacteriales order, our results are likely to have broad implications for understanding the regulation of analogous envelope stress responses in *Mtb* and other related organisms.

## Results

### A FACS-based screen for MOMP assembly factors

We selected the PorH porin as a model MOMP [14] because its C-terminus was previously shown to be detectable at the cell surface [15]. To monitor PorH assembly in the MM and its surface exposure, a multicopy plasmid encoding PorH with a 6x-His tag on the C-terminus (PorH-His) was constructed. This tagged variant of the porin was as functional as a corresponding untagged *porH* construct in complementing the kanamycin resistance phenotype of Δ*porH* cells (**S1 Fig**). To test for surface exposure of the His tag, cells producing PorH-His were incubated with a commercially available anti-His antibody labeled with Alexa Fluor 647 and imaged using fluorescence microscopy. We observed strong signal at the cell surface, indicating that we can detect surface exposed PorH that has been assembled into the MM (**Fig 1A**). As a control, we also probed PorH-His surface exposure in cells in which *cmt1* has been replaced with a zeocin-resistance cassette (*cmt1::zeo*). In the absence of Cmt1, PorH-His cannot be O-mycoloylated, leading to a dramatic decrease in PorH levels in the MM [7–9, 13]. As anticipated, *cmt1::zeo* cells expressing PorH-His showed no surface fluorescence (**Fig 1A**). We conclude that the PorH-His fusion provides a suitable probe to track MOMP assembly in the MM.

To screen for mutants that impair PorH assembly in the MM, the *porH-His* encoding plasmid was transformed into a previously constructed high-density transposon library [17]. Pooled transformants were then grown to mid-log, incubated with the fluorescent anti-His antibody, and sorted into "high fluorescent" and "low fluorescent" populations by FACS (**Fig 1B**). Wild-type and *cmt1::zeo* cells harboring the PorH-His plasmid served as controls to define the high fluorescent and low fluorescent gates for sorting, respectively (**Fig 1B**). Following collection of the high fluorescent (~94% of cells) and low fluorescent (~6% of cells) populations, the genomic DNA of both populations was isolated along with that of the unsorted library. The transposon insertion profiles in each population were then determined by transposon-sequencing (Tn-seq) [18]. We focused on genes that were enriched for insertions in the low fluorescent relative to the high fluorescent population because they were likely to encode factors required for the proper assembly of PorH-His at the cell surface (**Fig 1C and S1 Table**). As an indication that the screen was working as intended, the *cmt1* (*cgp_0413*) gene was identified as one of the top hits required for PorH-His surface exposure (**Fig 1C and S1 Table**).

### The σ$^D$ envelope stress response is required for proper MOMP surface exposure

Other top hits in the screen in addition to *cmt1* were *sigD* (*cgp_0696*), *marP (cgp_0356)*, and *rip1 (cgp_2207)* (**Fig 1C and 1D, and S1 Table**). The *sigD* gene encodes an alternative ECF sigma factor that is thought to control an envelope stress response (**Fig 2A**) [19–21]. It is conserved across the Mycobacteriales order along with its membrane-anchored anti-sigma factor

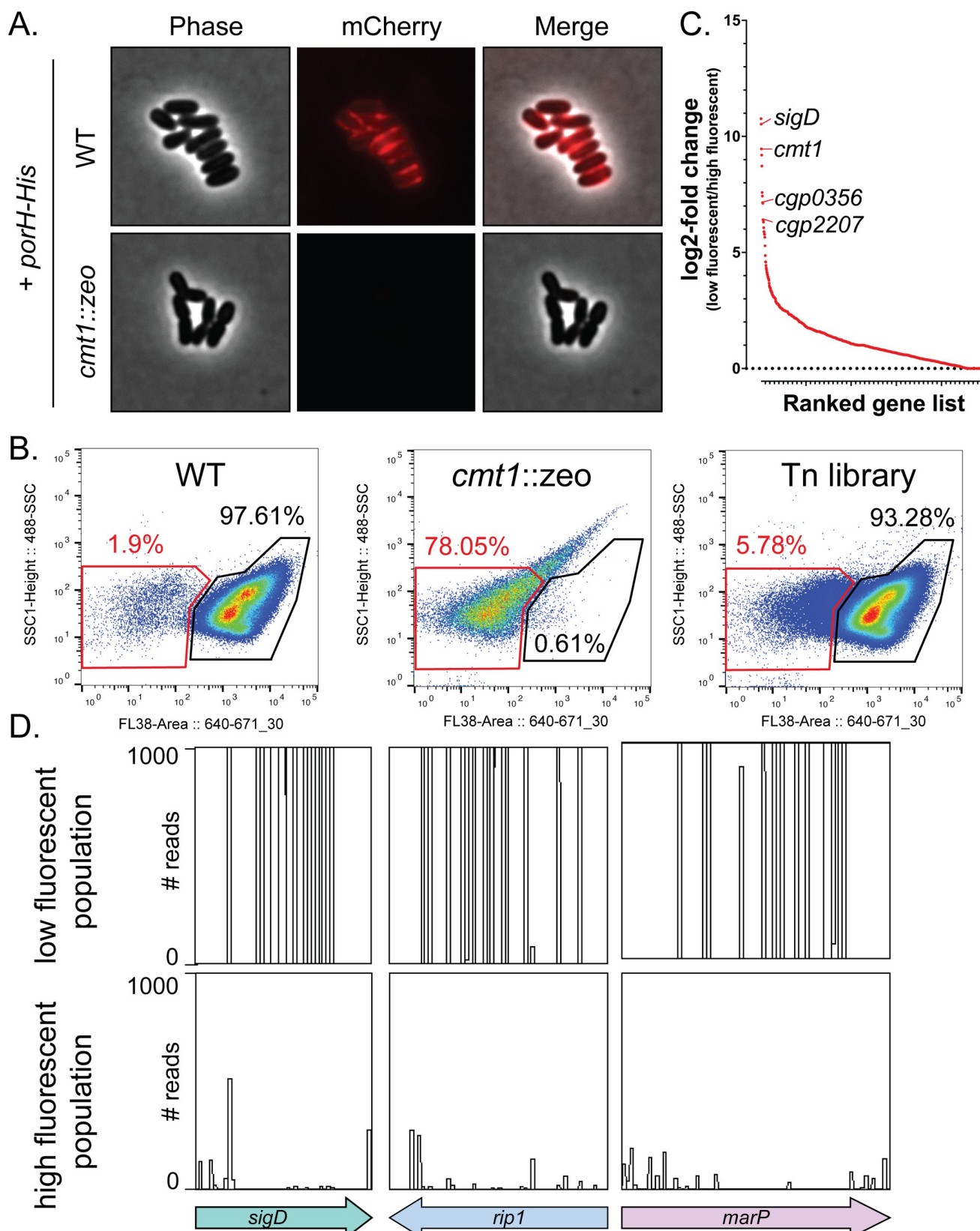

**Fig 1. FACS-based Tn-seq screen for MOMP assembly factors. (A)** Wild-type (WT) or *cmt1::zeo* cells expressing PorH-His were stained with an anti-His Alexa Fluor 647 antibody and visualized by fluorescence microscopy. **(B)** FACS plots of WT cells expressing PorH-His (high fluorescent gate), *cmt1:: zeo* cells expressing PorH-His (low fluorescent gate), and the high-density Tn library all expressing PorH-His. Cells were grown until-mid-log, stained with anti-6x His Alexa Fluor 647, and washed three times in 1X PBS. WT and *cmt1::zeo* cells were used to define the gates for sorting. Gates represent approximations and percentages represent the proportion of the total cell population that are within the experimentally sorted gates. **(C)** Ranked gene plot of log$_2$-fold value ratios of transposon insertions enriched in the low fluorescent population versus high fluorescent population. Each dot represents an individual gene. **(D)** Transposon insertion profiles of *sigD*, *marP*, and *rip1*. Each line signifies a site of transposon insertion, and the height of the line represents the number of reads. Data visualization made using Artemis [16].

RsdA [22–24]. Similar to the σ<sup>E</sup> envelope stress response in *E. coli*, activation of the σ<sup>D</sup> system is thought to involve regulated intramembrane proteolysis (RIP) of the anti-sigma factor by site-1 and site-2 proteases [23,24]. In *Mtb*, the RseP family member Rip1 functions as the site-2 protease for several anti-sigma factors, including RsdA [23,24]. The site-1 protease for the σ<sup>D</sup>

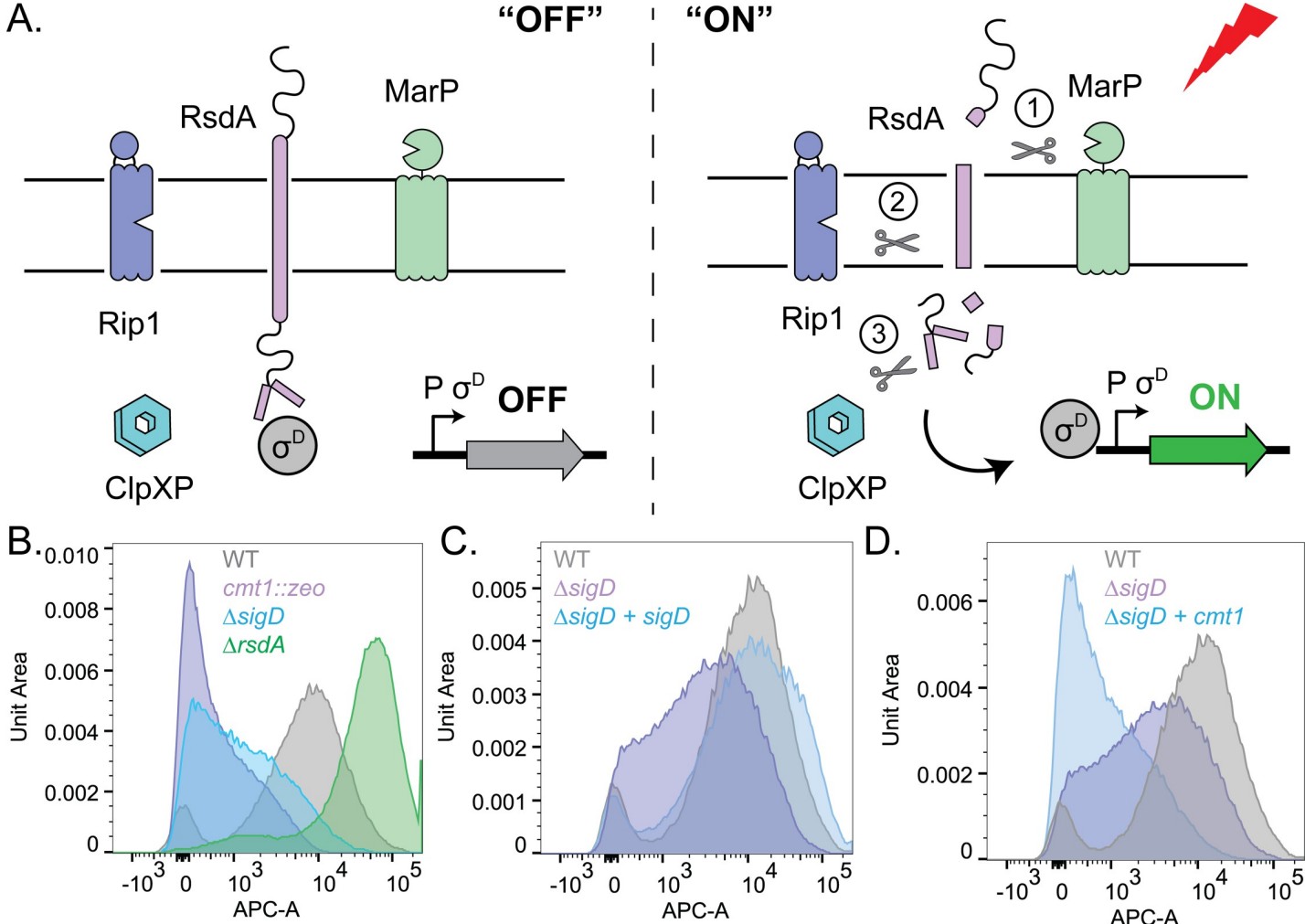

**Fig 2. The σ<sup>D</sup> pathway is required for MOMP surface exposure. (A)** Model of proposed σ<sup>D</sup> envelope stress response activation pathway. Under basal conditions, RsdA sequesters σ<sup>D</sup> at the inner membrane and prevents expression of the σ<sup>D</sup> regulon. In response to a stress signal, MarP, Rip1, and ClpXP sequentially cleave RsdA to release σ<sup>D</sup> to activate the σ<sup>D</sup> regulon. **(B-D)** Flow cytometry of WT cells or σ<sup>D</sup> pathway mutants expressing PorH-His. Cells were grown to mid-log and stained with anti-His Alexa Fluor 647. A representative replicate is displayed as a histogram and scaled using unit area. **(C)** An ectopic complement of *sigD* was expressed from a P$_{sod}$-riboE1 promoter in cells lacking the native copy of *sigD* expressing PorH-His. Cells were grown with 1mM theophylline for *sigD* induction prior to staining and analysis. **(D)** An ectopic complement of *cmt1* was expressed from a P$_{sod}$-riboE1 in *sigD* null cells expressing PorH-His. Cells were grown-until mid-log with 1mM theophylline for induction of *cmt1* prior to staining and analysis.

response has not been identified. However, the *marP* gene that was a hit in the screen along with *sigD* and *rip1* is conserved in *Mtb* and encodes a homolog of the *E. coli* site-1 protease DegS. MarP is therefore an attractive candidate for the site-1 protease of the σ<sup>D</sup> response. Thus, the screen for factors involved in PorH surface display identified many of the components of the σ<sup>D</sup> signaling pathway, suggesting that this regulatory system is required for proper MOMP assembly in the MM.

To confirm the role of the σ<sup>D</sup> response in PorH-His transport, we constructed mutants with deletions of *sigD* and *rsdA*. The PorH-His plasmid was transformed into these deletion backgrounds and cells were incubated with the fluorescent anti-His antibody. Surface exposure of PorH-His was then analyzed by flow cytometry and compared to wild-type (high fluorescent) or *cmt1*::*zeo* cells (low fluorescent). The *sigD* mutant exhibited an intermediate PorH-His transport phenotype in which fluorescence of the antibody treated cells had labeling intensities between the wild-type and *cmt1*::*zeo* cell populations (**Fig 2B**). This phenotype was complemented by ectopic expression of *sigD* (**Fig 2C**). Conversely, cells deleted of the anti-sigma factor *rsdA*, which causes constitutive σ<sup>D</sup> pathway activity, showed enhanced PorH-His surface exposure as indicated by their labeling profile shifting to greater intensities than wild-type cells (**Fig 2B**).

As *porH* and *cmt1* are both in the σ<sup>D</sup> regulon [22,25], we investigated if the impaired surface exposure of PorH-His in Δ*sigD* cells was due to reduced expression of these genes. We expressed a second copy of the *cmt1* from an ectopic location in the *sigD* deletion strain to increase Cmt1 levels. Surprisingly, when the second copy of *cmt1* was expressed at levels that correct the phenotype of *cmt1*::*zeo* cells (**S2 Fig**), there was no restoration of PorH-His surface exposure in the flow cytometry profile of Δ*sigD* cells (**Fig 2D**), but rather a strong decrease. Why overexpression of *cmt1* has this effect on Δ*sigD* cells remains unclear. We also tested whether the defect in PorH-His surface exposure in Δ*sigD* cells was due to changes in PorH expression by assaying protein levels using immunoblot analysis (**S3 Fig**). We detected native levels of ProtX (**S3A and S3B Fig**) that fluctuate with σ<sup>D</sup> activity, as ProtX is a member of the σ<sup>D</sup> regulon [22]. We also detected multiple proteoforms of Por-His that likely correspond to MOMPs at various stages of processing/modification, such as pre- and post-cleavage of a secretion signal and/or pre- and post-*O*-mycoloylation. All of these forms were drastically reduced in *cmt1*::*zeo* cells (**S3B and S3C Fig**), in agreement with previous studies (**S3B and S3C Fig**) [7,8,13]. We observed a slight but repeatable decrease in the upper proteoforms(s) of PorH-His upon deletion of *sigD* (**S3 Fig**). However, there appeared to be no changes to lower proteoform of PorH-His, which we believe corresponds to the mature protein, in Δ*sigD* cells. Thus, inactivation of the σ<sup>D</sup> pathway reduces the amount of potential precursor forms of PorH-His, but not mature protein. We conclude that the PorH-His surface exposure phenotype resulting from the inactivation of the σ<sup>D</sup> pathway is not due to lowered levels of Cmt1 or mature PorH, suggesting a role for members of the regulon in the processing of PorH and its assembly in the MM.

## Blocking arabinogalactan or mycolic acid synthesis activates the σ<sup>D</sup> pathway

Previous studies in *Mtb* and *Cglu* observed elevated expression of specific genes in the σ<sup>D</sup> regulon upon treatment with isoniazid and phenol [26,27], suggesting that defects in mycolic acid activates the pathway. To further investigate the signals that activate the σ<sup>D</sup> pathway, we constructed a reporter that fused the promoter of the σ<sup>D</sup>-controlled gene *cgp_2320* [22] to *lacZ* (P<sub>*cgp_2320*</sub>::*lacZ*). In wild-type cells carrying the reporter, β-galactosidase activity was comparable to that observed in Δ*sigD*, Δ*marP*, and Δ*rip1* cells, indicating that *cgp_2320* promoter is only weakly active if at all in unperturbed cells (**Fig 3A**). Importantly, deletion of the anti-

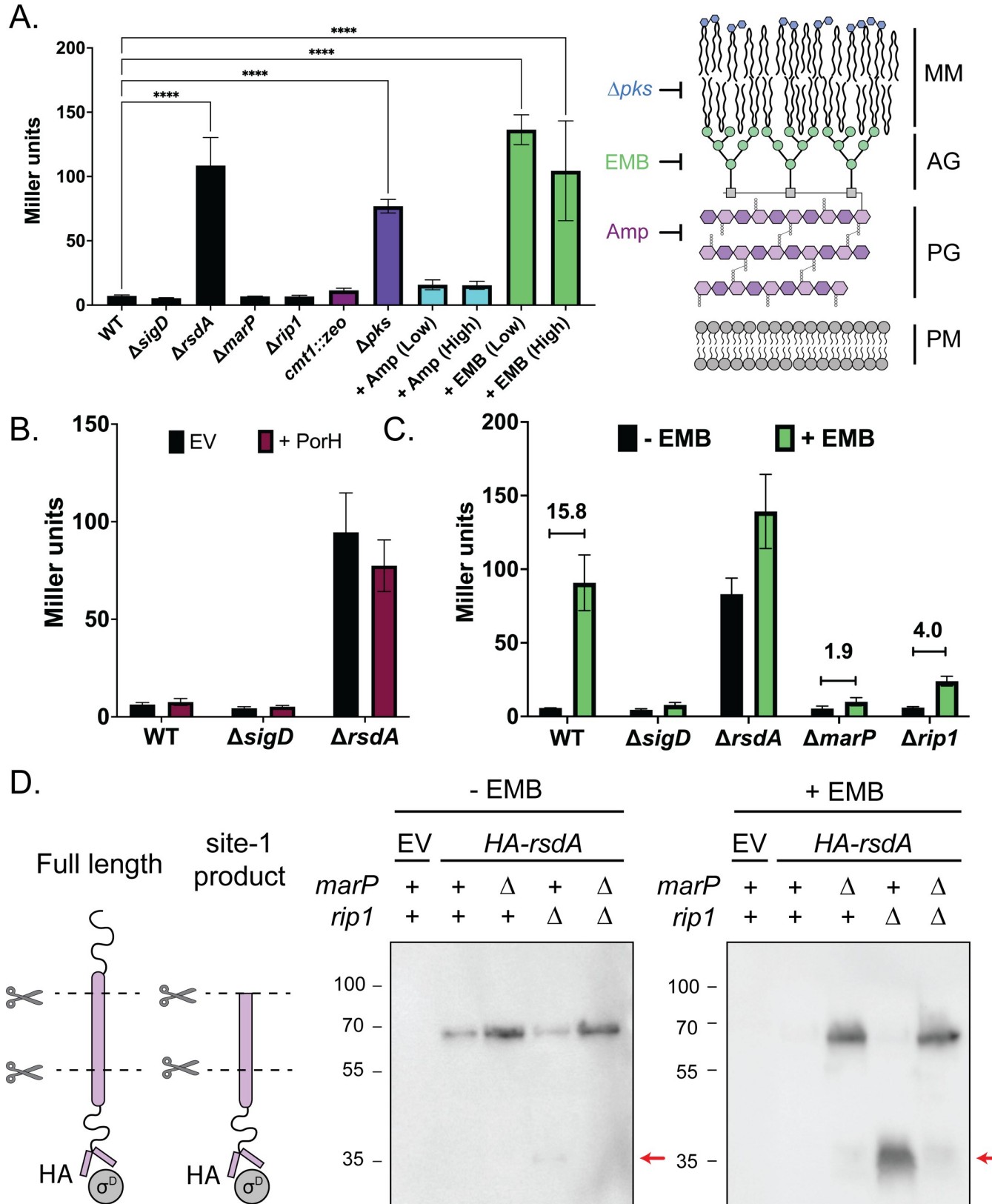

**Fig 3. The σ<sup>D</sup> pathway is activated by envelope stress in a MarP- and Rip1-dependent manner.** β-galactosidase assays were performed in biological triplicate. The average activity is graphed with the error bar representing standard deviation. **(A)** β-galactosidase assay of a panel (left) of treatments or genetic

backgrounds that target different aspects of the Mycobacteriales cell envelope (right). Cultures were grown, when indicated, with either 0.25x MIC ("low", 1.25µg/mL) or 0.5x MIC ("high", 2.5µg/mL) of the indicated antibiotic. Significance was determined using one-way ANOVA comparing to WT as a control (*** $p < 0.0001$). **(B)** β-galactosidase assay of the indicated strains carrying either an empty vector (EV) or plasmid expressing ectopic PorH-His from a P$_{tac}$ promoter induced with 1mM IPTG. **(C)** σ$^D$ pathway mutants were grown overnight in the presence (green) or absence (black) of 0.25x MIC of EMB (1.25µg/mL) and σ$^D$ reporter activity was measured by β-galactosidase assay. The displayed numbers denote fold-change in mean reporter activity between the–EMB and + EMB conditions indicated. **(D)** Investigation of HA-RsdA cleavage under stress conditions (left). Immunoblot analysis (right) of σ$^D$ pathway mutants lacking native *rsdA* with an ectopic HA-tagged complement expressed from a P$_{tac}$ promoter (no induction). Cells were grown overnight in the presence or absence of 0.25x MIC (1.25µg/mL) EMB. A strain carrying an empty vector (EV) served as a control.

sigma factor *rsdA* resulted in elevated levels of β-galactosidase activity, indicating that the P$_{cgp\_2320}$::*lacZ* reporter is sensitive to changes in σ$^D$ regulation.

To identify conditions that activate the σ$^D$ pathway, we exposed the reporter strain to a variety of chemical treatments and genetic perturbations that disrupt different aspects of cell envelope biogenesis (**Fig 3A**). Both ethambutol (EMB) treatment to disrupt AG synthesis and deletion of *pks*, a homolog of *Mtb pks13* that encodes an enzyme required for mycolic acid synthesis, resulted in strong induction of the reporter (19.1-fold and 10.8-fold increases, respectively for EMB and Δ*pks*) (**Figs 3A and S4**). No effect was observed upon treatment with ampicillin (Amp) to disrupt PG synthesis or deletion of *cmt1* to block protein mycoloylation (**Fig 3A**). To further test perturbations to PG biogenesis, reporter activity was monitored in strains lacking the SEDS-type PG synthase RodA or either of the two class A PBP-type PG synthases (PBP1a and PBP1b encoded by *ponA* and *ponB*, respectively), but no induction was observed (**S4 Fig**). Given that the analogous *E. coli* σ$^E$ response is induced by unfolded outer membrane proteins that can be artificially generated by overproducing them, we also tested the effect of *porH* overexpression (**S5 Fig**) on reporter activity. No induction was observed when the MOMP was overproduced (**Fig 3B**). We conclude that the σ$^D$ pathway is responsive to defects in the biogenesis of the AG layer and mycolic acids, but not defects in PG synthesis or MOMP assembly.

## MarP and Rip1 are required for RsdA cleavage and σ$^D$ pathway activation

With conditions that induce the σ$^D$ response identified, we next investigated the role of the MarP and Rip1 proteases in pathway activation. In other characterized ECF sigma factor systems that employ RIP, blocking proteolysis of the anti-sigma factor through deletion of the site-1 or site-2 proteases prevents sigma factor activation. We therefore monitored the P$_{cgp\_2320}$::*lacZ* reporter in Δ*marP*, and Δ*rip1* cells treated with EMB. These strains exhibited only 1.9-fold or 4-fold activation upon EMB treatment as compared to the 15-fold increase in σ$^D$ activity after wild-type cells were treated with EMB (**Fig 3C**). Similarly, deletion of *marP* and *rip1* strongly reduced reporter activity in cells deleted for *pks* compared to wild-type cells (**S6A Fig**).

In addition to the σ$^D$ transcriptional reporter, we also monitored processing of the RsdA anti-sigma factor in wild-type cells and mutants lacking MarP and/or Rip1. For these experiments, a functional N-terminal HA-tagged version of RsdA (HA-RsdA) (**S7 Fig**) was produced as the sole source of RsdA. Importantly, the gene encoding HA-RsdA was expressed under the control of a σ$^D$-independent promoter. Immunoblotting with anti-HA antibodies detected a band of the expected size of full-length HA-RsdA in untreated wild-type cells (**Fig 3D**). This band was undetectable following EMB treatment, consistent with the activation of proteolytic processing (**Fig 3D**). In Δ*marP* cells, elevated levels of HA-RsdA were detected in untreated cells, and these levels were unaffected by EMB treatment, indicating that proteolytic processing is blocked (**Fig 3D**). By contrast, levels of full-length HA-RsdA were similar to wild-type in untreated cells lacking Rip1, but a faint band of ~35kDa was specifically detected in this

background (**Fig 3D**). This ~35kDa HA-RsdA fragment accumulated to high levels following EMB treatment while the full-length protein was no longer detectable (**Fig 3D**). Results in Δ*marP* Δ*rip1* double mutant cells mirrored those of the Δ*marP* single mutant (**Fig 3D**). Thus, loss of MarP appears to block HA-RsdA processing completely whereas Rip1 inactivation appears to block the processing of a smaller HA-RsdA fragment produced by an initial cleavage step. These results combined with those using the transcriptional reporter support a model in which MarP functions as the site-1 protease and Rip1 as the site-2 protease for the RsdA cleavage reactions that release and activate σ$^D$.

## Disruption of primary arabinan chain formation activates the σ$^D$ response

EMB inhibits polymerization of the arabinan chains of AG from the precursor decaprenyl-arabinose (DPA) [28–32]. Thus, EMB treatment results in the formation of the galactan chain of the AG without its arabinan branches where mycolic acids are attached. Another consequence of EMB treatment is the accumulation of the DPA donor in the outer leaflet of the inner membrane [32], which would sequester decaprenyl lipid carrier needed for the biogenesis of other surface glycopolymers like PG and other lipoglycans [33, 34]. To investigate whether it is the loss of the arabinan chain or the accumulation of DPA that activates the σ$^D$ response, we monitored the activity of the P$_{cgp\_2320}$::*lacZ* reporter in a collection of mutants blocked at various stages of arabinan chain biogenesis (**Fig 4B, 4C, 4E and 4F**). Such mutants are possible to construct in *Cglu* because the MM and arabinan branches of the AG are not essential in this organisms unlike other commonly studied members of Mycobacteriales like *Msmeg* and *Mtb* where they are required for growth.

Synthesis of the DPA precursor of the arabinan chain occurs on the cytoplasmic face of the inner membrane (**Fig 4A**). UbiA (Cgp_3189) loads phosphoribose-diphosphate onto the decaprenyl carrier to form decaprenylphosphoryl-5-phosphoribose (DPPR) [35,36]. DPPR is then converted to decaprenylphosphoryl-5-ribose (DPR) by Cgp_3190/Cgp_3193 [37]. The essential enzyme DprE1 (Cgp_0238) and redundant enzymes DprE2/Cgp_1680 then convert DPR into DPA [38,39], which is then flipped to the periplasmic face of the inner membrane by an unidentified transporter. After the DPA donor is flipped, a set of arabinofuranosyltransferases add arabinan residues to the galactan core (**Fig 4D**) [40–42]. First, AftA (Cgp_0236) adds the priming arabinan residue [40] and Emb, the target of the EMB drug [29,30,43], polymerizes the arabinan residues into a linear chain [28]. AftC (Cgp_2077) initiates branching of the growing chain [42] and AftD (Cgp_3161) elongates these branches [41]. Finally, AftB (Cgp_3187) caps the non-reducing ends of the growing AG molecule, generating the sites for modification with mycolic acid to form the inner leaflet of the MM [44,45].

Blocking DPA synthesis either by deletion of *ubiA* or treatment with the DprE1 inhibitor TCA1 [46] resulted in strong σ$^D$ activation (**Fig 4B and 4C**). Co-treatment of cells with both EMB and TCA-1 resulted in reporter activity that was equivalent to either treatment alone (**Fig 4C**). As expected, disruption of *emb* resulted in σ$^D$ activity as did inactivation of *aftA* to prevent arabinan chain initiation (**Fig 4E and 4F**). By contrast, inactivation of the arabinofuranosyltransferases that decorate the primary arabinan chain (*aftC,D, or B*) did not result in a significant increase in σ$^D$ activity (**Fig 4F**). Thus, these results suggest that the σ$^D$ signaling pathway does not respond to a reduction in the free pool of the decaprenyl lipid carrier or to elaborations in AG branch chaining but instead responds to the loss of the primary arabinan chain itself.

In addition to their use in forming the arabinan chain of the AG layer, DPA precursors are transferred to an inner membrane lipoglycan called lipomannan (LM) to form lipoarabinomannan (LAM) [47]. The arabinosyltransferase(s) responsible for LM/LAM formation are

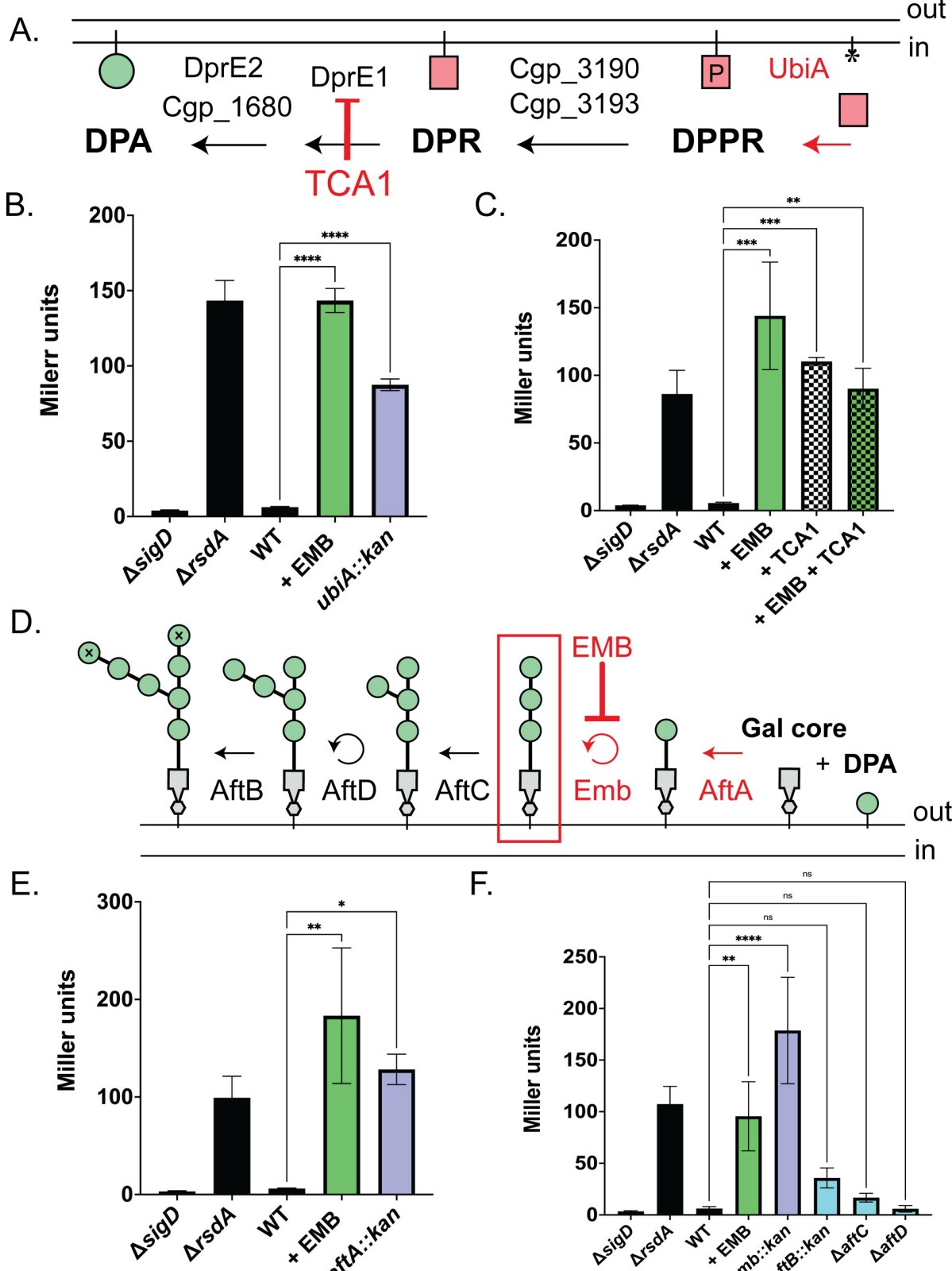

**Fig 4. The σ^D pathway responds to defects in the primary polymerized arabinan formation.** For β-galactosidase assays, graphs represent the average reporter activity with standard deviation of three biological replicates. **(A)** Pathway for DPA biosynthesis. The membrane

displayed is the inner membrane. The red square denotes phosphoribose, the asterisk is empty decaprenyl carrier, and the green circle represents arabinan residues. DPPR = decaprenylphosphoryl-5-phosphoribose, DPR = decaprenylphosphoryl-ribose, DPA = decaprenylphosphoryl-D-arabinose. **(B-C, E-F)** Measurement of σ<sup>D</sup> reporter activity by β-galactosidase assay in the indicated strains. Significance was determined using one-way ANOVA comparing to WT as a control (**** $p < 0.0001$, *** $p < 0.001$, ** $p < 0.01$, * $p < 0.05$, ns = not significant) **(C)** β-galactosidase activity in cells grown overnight in the absence (black) or in the presence of either of 0.25x MIC EMB (1.25μg/mL) (green) or 0.25x MIC TCA1 (1.25μg/mL) (checked) or both (green/checked). **(D)** Model of AG biosynthesis on the outer leaflet of the inner membrane. The gray polygon is galactan core, the green circles are arabinan residues, and the green circles with "X" are terminal arabinan residues. Chemical treatments or genetic disruptions that activate the σ<sup>D</sup> pathway in red. The hypothesized molecule that is sensed is boxed in red.

currently unknown. LMs and LAMs play critical role(s) in the cell envelope as evidenced by the fitness defects upon deletion of LAM biosynthetic enzymes [48–51]. In *Mtb* and *Msmeg*, the Emb homolog EmbC catalyzes the transfer of arabinan residues to LAM and is inhibited by EMB [52]. Thus, it is formally possible that activation of the σ<sup>D</sup> pathway by EMB treatment is due to defective LAM biosynthesis. To investigate this possibility, we monitored σ<sup>D</sup> activity in strains deleted for the genes encoding the mannosyltransferases MptA (Cgp_2385*)* [50], MptB (Cgp_1766*)* [49], or MptC (Cgp_2393) [51] needed for LM and LAM biogenesis. σ<sup>D</sup> activity was unaffected in these mutants, indicating that defective LM/LAM production is not a signal sensed by this stress response pathway (**S8 Fig**). Based on the overall analysis of the AG and LM/LAM pathways, we conclude that the σ<sup>D</sup> response is induced by inhibiting the biogenesis of the primary arabinan chain of the AG layer. Accumulation of DPA, defects in addition of branches to the arabinan chain, or the inhibition of LM/LAM biogenesis do not appear to be monitored by the response.

## Evidence that the σ<sup>D</sup> pathway monitors levels of mature mycolic acids in the inner membrane

A common feature of the σ<sup>D</sup> activation conditions we identified, EMB treatment and deletion of *pks*, is that they both prevent proper MM biogenesis [44]. EMB inhibits the polymerization of the arabinan chains of AG used for the covalent attachment of mycolic acids to form the MM whereas inactivation of Pks completely blocks the synthesis of mycolic acids to prevent MM biogenesis. Thus, blocking either an early or late step required for MM biogenesis results in σ<sup>D</sup> activation. We were therefore curious whether blocking any step in mycolic acid synthesis or transport activates the σ<sup>D</sup> response.

Mycolic acid biosynthesis (**Fig 5A**) begins with the formation of long chain fatty acids by fatty acid synthase I enzymes to generate acyl-CoA molecules that will become the α branch and the meromycolate chains of mycolic acids [1,53]. These molecules are activated by either the acyl-CoA carboxylase PccB (Cgp_3177, homolog of *Mtb* AccD4) [54] or the acyl-AMP ligase FadD2 (Cgp_3179, homolog of *Mtb* FadD32) [55]. The two activated fatty acids are then condensed into a myolic acid molecule by Pks in a Claisen-like condensation reaction [56–58]. Pks activity requires 4'-phosphopantetheinylation by the enzyme PptA (Cgp_2171, homolog of *Mtb* PptT) [59]. Once formed, the mycolic acid molecule is linked to trehalose by Pks [56]. Trehalose can be imported into the cell by a dedicated transporter complex [60] or synthesized in the cytoplasm from glucose by either the OtsAB [61], TreS, or TreYZ biosynthetic pathways [62,63]. The trehalose mycolate is then reduced by CmrA (Cgp_2717) to form trehalose mono-corynomycolate (TMCM) in the inner leaflet of the inner membrane [64,65]. TMCM is further modified by the proteins MmpA (Cgp_3165) and TmaT (Cgp_3163), and these modifications appear to be required for proper TMCM transport [66,67]. AhfA (Cgp_0475) has also been shown to be required for TMCM synthesis in *Cglu*, but its role in the process remains unclear [68].

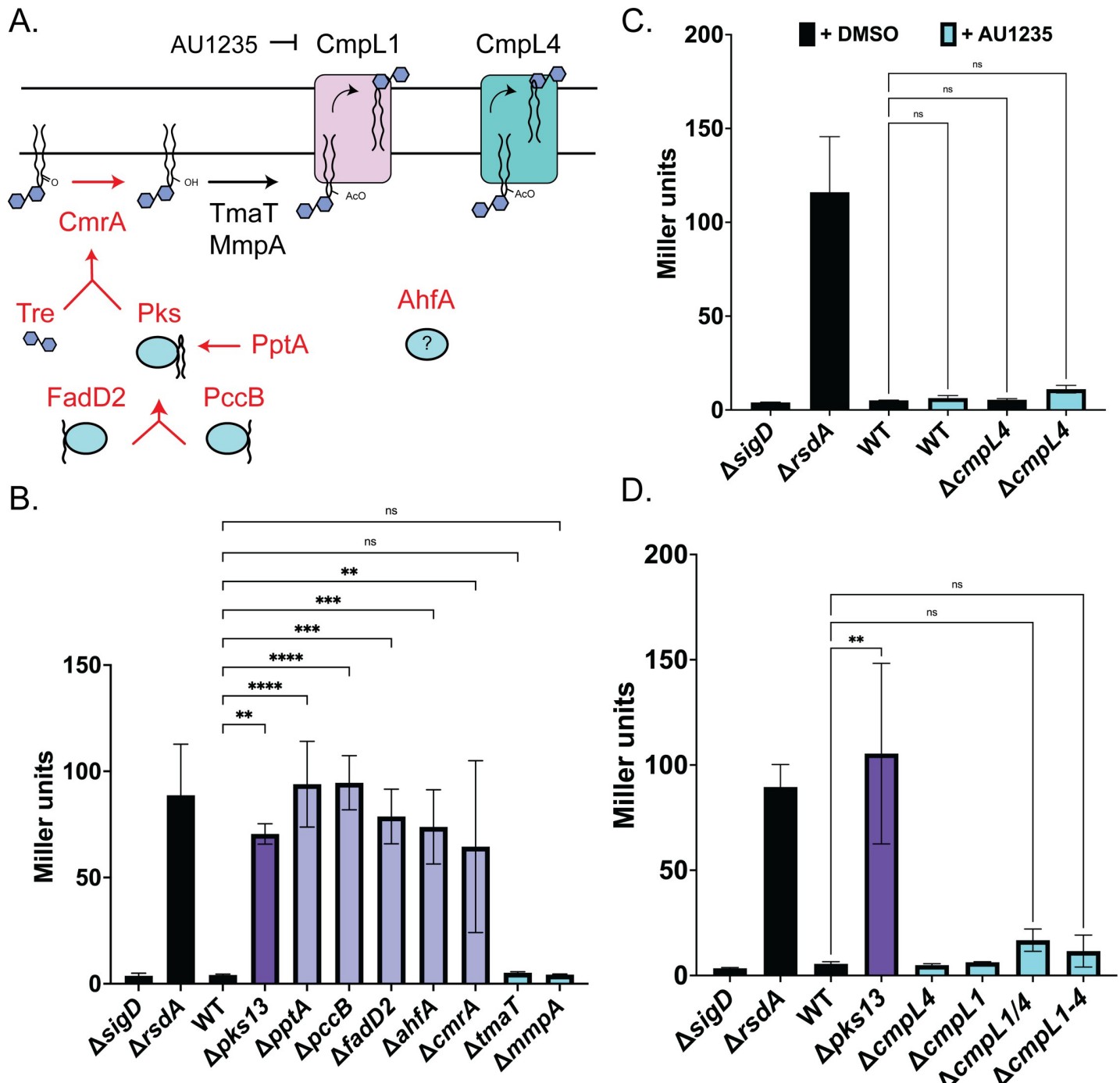

**Fig 5. The σ^D pathway responds to the absence of mature mycolic acid in the inner membrane. (A)** Pathway for mycolic acid biosynthesis. Proteins in red denote steps that activate the σ^D pathway when deleted. The molecule boxed in red is hypothesized to be sensed by the σ^D pathway. **(B-D)** Data shown is the average of three replicates and error bars show standard deviation. Significance was determined using one-way ANOVA comparing to WT as a control (** $p < 0.01$, *** $p < 0.001$, **** $p < 0.0001$, ns = not significant). **(B)** Reporter analysis by β-galactosidase assay in a panel of mycolic acid biosynthetic mutants. **(C)** β-galactosidase assay of cells treated with the vehicle control (DMSO) or 0.025mM AU1235 [67] in control strains or strains lacking *cmpL4*. **(D)** σ^D reporter activity as assayed by β-galactosidase activity in the indicated *cmpL* mycolic acid transport mutants.

Deletion of genes encoding the enzymes involved in the early steps of mycolic acid synthesis (*pccB*, *fadD2*, *pptA*) all induced the P$_{cgp\_2320}$::*lacZ* reporter to levels comparable to the deletion of *pks* (**Fig 5B**). Deletion of *ahfA* also induced the response, further supporting its connection with mycolic acid synthesis (**Fig 5B**). Inactivation of the first enzymes in two of the three major trehalose biosynthetic pathways in *Cglu* [62,69], *otsA* (*cgp\_2907*) and *treY* (*cgp\_2323*), individually did not induce the reporter (**S9 Fig**). However, a double mutant lacking both enzymes, which has been shown to deplete trehalose from the cell wall [62], resulted in activation of the σ$^D$ reporter (**S9 Fig**). Inactivation of the trehalose-mycolate reducing enzyme CmrA also activated the response but defects in the additional modifying enzymes (Δ*tmaT* and Δ*mmpA*) did not (**Fig 5B**). Thus, blocking the formation of TMCM prior to the final two modification steps triggers the σ$^D$ pathway.

We next tested whether blocking the transport of TMCM to the MM might also serve as an inducing signal for the σ$^D$ response. In *Mtb*, a single transporter called MmpL3 has been implicated in the transport of mycolic acids across the inner membrane [70–72]. In *Cglu*, two partially redundant transporters, CmpL1 (Cgp_3174) and CmpL4 (Cgp_0284), are thought to perform the same function. The MM is formed in mutants lacking a single transporter but not in a double mutant. Only CmpL1 is sensitive to the MmpL3 inhibitor AU1235 [73,74]. We therefore tested whether this compound would induce the σ$^D$ reporter in cells in which *cmpL4* was deleted where AU1235 should effectively block all remaining TMCM transport by CpmL1. Surprisingly, no induction was observed in Δ*cmpL4* cells treated with AU1235 (**Figs 5C and S10**). Similarly, no induction was observed in a mutant deleted for both *cmpL1* and *cmpL4* or a mutant deleted for all four genes in *Cglu* encoding MmpL-like proteins (**Fig 5D**). We therefore infer that the σ$^D$ pathway is not responding to defects in MM assembly per se in these instances but rather that it senses the lack of TMCM in the inner membrane.

## Discussion

The mechanisms required for proper cell envelope biogenesis in the Mycobacteriales order of bacteria remain poorly understood, including those involved in the transport of proteins to the MM and their assembly within this outer envelope layer. We therefore set out to identify factors involved in the assembly of MOMPs in the MM by screening for *Cglu* mutants with reduced levels of PorH exposed on their surface. Mutants inactivating the σ$^D$ envelope stress response were among the strongest hits in the screen. Although this result did not reveal a discrete set of components involved in MOMP assembly in the MM as we had hoped, it provided us with an opportunity to address outstanding questions related to the regulation of the σ$^D$ response.

## MarP is the site-1 protease of the σ$^D$ pathway

Prior work investigating the σ$^D$ response in mycobacteria revealed that RsdA is the anti-sigma factor and that the site-2 protease that helps release σ$^D$ from RsdA to activate it is Rip1 [23,24]. However, the identity of the site-1 protease that initiates the RIP cascade has remained unknown. Our results provide strong evidence that MarP serves this function. Mutants lacking MarP were hits in our screen along with those inactivated for *sigD* and *rip1* (**Fig 1D**). Cells deleted for *marP* are also defective in the activation of a σ$^D$ responsive promoter upon exposure to inducing conditions such as EMB treatment or the inactivation of *pks* (**Figs 3C and S6**). Furthermore, immunoblotting revealed that RsdA processing is blocked in Δ*marP* cells following EMB treatment (**Fig 3D**). Thus, the likely signaling cascade for σ$^D$ activation involves the sequential processing of RsdA by MarP and Rip1 followed by its final cleavage by ClpXP to release σ$^D$ to transcribe its regulon (**Fig 2A**) [75].

## A potential sensing mechanism monitoring mycolic acid in the inner membrane

To investigate the signals leading to induction of the σ<sup>D</sup> pathway, we tested its response to defects in MM assembly. Deletion of genes encoding early steps in the synthesis of mycolic acid preceding formation of mature trehalose monomycolate (TMCM) (Δ*fadD2*, Δ*pccB*, Δ*pks*, Δ*pptT*, Δ*otsA*Δ*treY*, and Δ*cmrA)* activated the σ<sup>D</sup> pathway (**Figs 5B and S9**). Surprisingly, however blocking the modification or transport of TMCM needed for MM assembly (Δ*tmaT*, Δ*mmpA*, Δ*cmpL1/4*, and AU1235 treatment) failed to stimulate the response (**Fig 5B–5D**). Thus, the RIP cascade leading to σ<sup>D</sup> activation appears to be held inactive as long as trehalose-linked mycolic acids are present in the inner membrane. This observation suggests a potential role for these lipids as repressors of RsdA processing, potentially through their association with the anti-sigma factor or the regulatory proteases. Notably, a previous study identified Rip1 as a potential target of *O*-mycoloylation due to the presence of a conserved *O*-mycoloylation motif in the transmembrane region of the protein [7]. Serine-serine signatures that are found in this *O*-mycoloylation motif are also present in both RsdA and MarP, suggesting an attractive potential mycolic acid sensing mechanism involving responsiveness to the lipidation status of one or more proteins in the RIP cascade.

## A second activation signal involving defects in AG biosynthesis?

In addition to defects in mycolic acid synthesis, inhibition of AG biosynthesis also serves as an inducer of the σ<sup>D</sup> response. Our results suggest that the molecule in the AG biogenesis pathway that is monitored by the σ<sup>D</sup> pathway is the primary arabinan chain of AG molecules. Indeed, both genetic backgrounds (*ubiA*::*kan*, *emb*::*kan*, and *aftA*::*kan*) and chemical perturbation (TCA1 and EMB treatment) that truncate AG molecules results in pathway activation (**Fig 4B and 4E**). We note, however, that the possibility remains that the *aftA*::*kan* mutation is polar on the downstream gene, *emb*. In contrast, inhibiting branching and terminal capping of AG did not activate the response (**Fig 4F**). Thus, disruption of MM biogenesis is not the ultimate inducer of the σ<sup>D</sup> response. One possible connection between the inducing properties of the mycolic acid and AG defects might be via a feedback mechanism through which defects in synthesis of the primary arabinan chain of AG causes the down-regulation of TMCM synthesis. However, prior studies have shown that although blocking AG biosynthesis via EMB treatment results in a decrease of cell-wall bound lipids, it does not decrease mycolic acid production. Rather, EMB treatment leads to unchanged or, in some studies, increased TMCM biosynthesis [76–80]. Thus, we believe that the mycolic acid and AG activating signals are distinct and that the σ<sup>D</sup> response monitors the status of multiple aspects of the envelope. How the σ<sup>D</sup> pathway might sense the AG defects is not clear, but an attractive possibility is that RsdA might use its long, disordered periplasmic domain of approximately 160 amino acids to detect AG defects in a similar manner to which the disordered domain of the anti-sigma factor RsgI in *Bacillus subtilis* senses cell wall defects to induce the σ<sup>I</sup> response [81]. Alternatively, activation may be induced by a buildup of galactan core in the inner membrane or possible side effects of EMB treatment and/or mutants defective in AG biogenesis on inner membrane integrity.

## Potential role of the σ<sup>D</sup> response in maintenance of envelope homeostasis

The regulons controlled by alternate sigma factors often directly address the defects that promote pathway activation to re-establish envelope homeostasis. The best characterized example of this homeostatic response is the σ<sup>E</sup> system in *E. coli*. Defects in outer membrane protein assembly and mislocalized lipopolysaccharide cause σ<sup>E</sup> activation [82,83]. Accordingly, the σ<sup>E</sup>

regulon includes genes encoding periplasmic chaperones, the outer membrane protein assembly machinery (BAM complex), and components of the lipopolysaccharide transport machinery (Lpt system) [84,85] to ameliorate the problems that stimulated induction of the response.

Previous studies have shown that the σ$^D$ regulon contains a number of genes related to mycolic acid biosynthesis (*fadD2*, *pks*, *pccB*), cell wall homeostasis (*lppS*, putative glycosyltransferases), and MM integrity (*porA*, *porH*, mycoloyltransferases) [22,25,26]. Under conditions of reduced TMCM in the inner membrane that we have shown activate the σ$^D$ pathway, increased expression of mycolic acid biosynthetic genes may enable to cell to rapidly increase TMCM production to respond to a perceived deficit. Similarly, the alteration of peptidoglycan structure by increasing the level of the crosslinking enzyme LppS may circumvent defects in AG biosynthesis that promote σ$^D$ activation. Indeed, previous work has suggested that in the absence of normal peptidoglycan crosslinks AG is not properly ligated onto peptidoglycan [86]. We therefore propose that the σ$^D$ regulon, like the σ$^E$ response in *E. coli*, is similarly poised to fix problems in envelope assembly that promote its induction.

## A potential role for σ$^D$ regulon members in MOMP assembly

Multiple regulatory components of the propose σ$^D$ pathway were hits in our screen for MOMP assembly factors. Additionally, activation of the σ$^D$ response was found to alter the abundance of different proteoforms of PorH, which is suggestive of defects in proper MOMP processing or modification needed for assembly into the MM. Thus, just as the *E. coli* σ$^E$ regulon includes components of the BAM machinery needed for β-barrel protein assembly in the outer membrane, we think it is likely that the σ$^D$ regulon in *Cglu* includes components of the MOMP processing and assembly pathway that are yet to be identified. We anticipate that the MOMP surface exposure assay developed as a part of this study will be useful in testing this hypothesis and defining this important aspect of envelop assembly in corynebacterial and their relatives.

## Materials & methods

### Bacterial strains & growth conditions

*C. glutamicum* MB001, a prophage-free strain derived version from ATCC 13032, was used for all experiments [87]. *C. glutamicum* strains (**S2 Table**) were grown in brain heart infusion (BHI) medium (BD) or brain heart infusion medium supplemented with 9.1% sorbitol (BHIS) at 30˚C with aeration, as indicated. Ectopic expression constructs were induced either with 1mM theophylline when expressed from *attB*(P$_{sod}$-*riboE1*) [88,89] or relied on leaky expression (no IPTG) from an inducible P$_{tac}$ promoter when expressed from a pTGR5-derived replicating plasmid [90], as indicated. *E. coli* DH5α(λ$_{pir}$) cloning strain (NEB) was propagated at 37˚C in LB (1% tryptone, 0.5% yeast extract, 0.5% NaCl) with aeration unless harboring pCRD206 derivatives, in which cells were cultured at 30˚C. *C. glutamicum* strains were grown with 15μg/mL kanamycin, 3.5μg/mL chloramphenicol, or 150μg/mL zeocin, when appropriate. *E. coli* strains were grown in 25μg/mL kanamycin, 25μg/mL chloramphenicol, or 25μg/mL zeocin, when necessary. Chemical inhibitors were used at the following concentration: EMB high: 2.5μg/mL, EMB low: 1.25μg/mL, ampicillin high: 2.5μg/mL, ampicillin low: 1.25μg/mL, AU1235: 0.025mM, TCA1: 1.25μg/mL.

### Plasmid construction

Plasmids were constructed using isothermal assembly (ITA) (**S3 Table**) and transformed into *E. coli* DH5α(λ$_{pir}$) (NEB) competent cells by heat shock (42˚C for 40 seconds). Primers used in

plasmid construction are listed in **S4 Table** and were purchased from IDT or Genewiz. gBlocks were purchased from IDT using IDT's codon optimization tool for *Corynebacterium glutamicum*. To construct pCRD206 derivatives [91] to delete genes by allelic exchange, 500-750bp fragments encoding homology regions upstream and downstream of the gene of interest leaving a small scar fragment of the coding region to maintain the reading frame were assembled into the pCRD206 by ITA. pCRD206 derivatives [91] to disrupt genes by allelic exchange were constructed in the same manner, however a zeocin-resistance cassette was encoded within the scar fragment. The pEMH25 empty vector was constructed by replacing the *lacI/P$_{tac}$::eGFP* fragment from the published pTGR5 vector [90] with the P$_{sod}$ promoter and native *sod* RBS from the *C. glutamicum* genome using ITA. Subsequently, the kanamycin resistance cassette was replaced with a chloramphenicol resistance cassette using ITA and the region downstream of the promoter was deleted using site-directed mutagenesis using KLD enzyme mix (NEB). The pEMH120 empty vector was constructed from a pTGR5 [90] derivative that is chloramphenicol resistant by removing the native BamHI site, inserting a new BamHI site downstream of the P$_{tac}$ promoter, and deleting the *eGFP* open reading frame using site-directed mutagenesis with KLD enzyme mix (NEB). Ectopic complementation vectors were cloned onto pSEC1 (derivative of pK-PIM integrating vector, P$_{sod}$-riboE1, kanamycin-resistant or zeocin-resistant) [88,89] or pEMH120. All genes were amplified from MB001 gDNA. Tags (6x His or HA) were introduced using site-directed mutagenesis with KLD enzyme mix (NEB).

## Colony PCR in *C. glutamicum*

Colony PCR was performed using either Q5 (NEB) or Sapphire master mix (Fisher Scientific, Clonentech Labs) from cells grown on solid agar. For Q5 colony PCR, cells were lysed in 10μL of 2mg/mL lysozyme for 30 minutes at 37˚C. Lysis was completed by adding 10μL of nuclei lysis buffer (Promega) and diluting with 100μL of ddH$_2$O. PCR reactions were set up with 6.25μL of Q5 master mix, 5.25μL of ddH$_2$O, 0.625μL of forward and reverse primer (10μM), and 0.5μL of lysate. For Sapphire colony PCR, cells were lysed in 5μL of 1mg/mL lysozyme for 30 minutes at 37˚C. Lysis was completed by adding 5μL of nuclei lysis buffer (Promega) and 180μL of ddH$_2$O. PCR reactions were set up with 5μL of Sapphire master mix, 3.7μL of ddH$_2$O, 0.5μL of forward and reverse primer (10μM), and 0.3μL of lysate.

## Strain construction

*C. glutamicum* competent cells were prepared as previously described [17,92]. Briefly, strains were sub-cultured from an overnight culture into transformation media (BHI supplemented with 91g sorbitol, 0.1% Tween 80, 0.4g isoniazid, 25g glycine) and grown until in mid-log (approximately 3 hours at 30˚C or overnight at 18˚C). Cells were washed two times in 10% glycerol spinning at 3,000rpm and resuspended in 10% glycerol. Gene deletion or disruption was performed using allelic exchange via *sacB* counterselection with pCRD206 derivative plasmids [91] or by recombineering, as described in more detail below. Integration of pK-PIM derivatives (pSEC1 variants) [89] was validated by colony PCR.

## Recombineering in *C. glutamicum*

**Plasmid construction.** Recombineering plasmids (pEWL54 and pEWL103) were constructed by cloning a gBlock encoding a codon-optimized single-strand annealing protein (SSAP) and cognate single-strand binding protein (SSB) from *Troponema socranskii* prophage with an internal RBS generated by a publicly available RBS calculator (https://salislab.net/software/predict_rbs_calculator) into either a pTGR5-derivative [90] (pEWL54) or a pCRD206-derivative [91] (pEWL103). The selection identifying the specific SSAP/SSB gene

pair for recombineering will be described elsewhere. The recombineering cassette template plasmid (pEWL74) was constructed from a gBlocks containing a multiple cloning site containing a kanamycin resistance cassette amplified from a pK-PIM derivative [89] flanked by LoxP71 and LoxP67 sites. The Cre resolving plasmids (pEWL73 and pEWL89) contain a *cre* recombinase driven by a P*sod* promoter. pEWL73 is a chloramphenicol resistant pTGR5 derivative [90] in which *repA* has been swapped for a temperature-sensitive *repA* from pEC-XK99E [93] (temperature sensitive above 34˚C). pEWL89 is a pCRD206 derivative [91] that is apramycin resistant (temperature sensitive above 25˚C).

**Recombineering induction conditions.**   To induce expression of the SSAP/SSB cognate pair from pEWL54, cells harboring the plasmid were grown in BHIS supplemented with chloramphenicol overnight at 30˚C. Overnight cultures were diluted 1:50 in transformation media at 30˚C until the culture had reached mid-log (~3 hours). Expression of the SSAP/SSB proteins was then induced by adding IPTG to a final concentration of 1mM and growing at 30˚C for another two hours. Competent cells were then prepared as described above. To prepare recombineering cells from cells harboring pEWL103, a saturated culture was grown in BHIS containing apramycin at 25˚C overnight. Cells were diluted 1:100 and grown in transformation media at 25˚C for 4 hours. The SSAP/SSB pair was then induced by adding IPTG and theophylline to final concentrations of 1mM each and growing for another 4 hours at 25˚C. Competent cells were then prepared as described above.

**Linear dsDNA kanamycin resistance cassette preparation.**   Linear dsDNA cassettes were constructed using a pair of 70-mer oligonucleotides. The 70-mer oligonucleotides were used to amplify the kanamycin resistance cassette flanked by LoxP66 and LoxP71 sites located on pEWL74. Each 70-mer oligonucleotide contained 20bp of homology to the cassette and 50bp of homology to the genomic region of interest.

**pEWL54/pEWL103 electroporation, screening, and curing.**   Induced competent cells carrying pEWL54 or pEWL103 were transformed with 500ng of purified linear dsDNA following the conditions described above. Cells were recovered for 2 hours at 30˚C and the entire transformation was plated on BHIS agar containing kanamycin to select for recombinants. Recombinants were confirmed using colony PCR. Plating the transformation at 30˚C is sufficient for curing of pEWL103, where appropriate. Validated recombinants carrying pEWL54 were grown overnight in BHIS at 30˚C and streaked or plated onto BHIS agar without antibiotic selection. These colonies were then patched onto BHIS + kanamycin and BHIS + chloramphenicol to identify isolates in which pEWL54 has been cured.

**Cassette curing with pEWL73/pEWL89.**   Electrocompetent cells of recombineered strains were prepared as described above. pEWL73 or pEWL89 was then transformed by electroporation and recovered/plated at 30˚C on chloramphenicol for selection in the case of pEWL73 or recovered/plated at 25˚C on apramycin for pEWL89. Transformants were streaked on BHIS at 30˚C and then patched to ensure that the kanamycin resistance cassette had been removed. In the case of pEWL89, patching at 30˚C was sufficient to cure the plasmid. In the case of cells harboring pEWL73, cells were sub-cultured 1:100 in BHIS, grown for 3 hours at 37˚C, and plated onto BHIS. Colonies were then patched to BHIS and BHIS + chloramphenicol to confirm that pEWL73 had been cured. Excision of the kanamycin resistance cassette was confirmed by colony PCR.

## Gene deletions in *C. glutamicum* by allelic exchange

Allelic exchange via *sacB* counterselection was performed as previously described [91]. The pCRD206 derivative was transformed into the appropriate strain via electroporation and recovered at 25˚C on BHI(S) Kan15 to allow for plasmid replication. Transformants were

restreaked to BHI(S) Kan15 at 30˚C to isolate integrants. Integrants were grown overnight at 25˚C in 1mL BHI(S) and plated on BHI + 10% sucrose agar at 30˚C to select against *sacB* encoded on the pCRD206 vector. Candidates were further screened by colony PCR with primers that anneal upstream and downstream of the gene of interest to identify isolates harboring the deletion.

## Tn-seq screen and fluorescence activated cell sorting (FACS)

The high-density transposon library was previously generated [17]. Tn library competent cells were prepared as previously described [17,92]. pEMH27 was transformed into 10 aliquots of Tn library competent cells by electroporation, as previously described [17,92]. Transformed cells were recovered for 1 hour at 30˚C and plated undiluted on BHI Cam3.5 at 30˚C. Plates were grown for 24 hours at 30˚C and 16 hours at 25˚C. Cells were scraped and frozen in 15% glycerol. A frozen aliquot of the Tn library harboring pEMH27 ($OD_{600}$ = 28) was thawed on ice. Approximately $7 \times 10^7$ cells (5μL) of the thawed library was used to sub-culture 5mL of BHIS Cam3.5 and grown at 30˚C for 4.5 hours. Sub-cultures of control strains (MB001 pEMH27 and *cmt1*::*zeo* pEMH27) were made by inoculating 50μL of an overnight culture in 5mL BHIS Cam3.5 and grown at 30˚C for 4.5 hours. After growth, and $OD_{600}$ equivalent to 0.1 was pelleted for 3 minutes at 12,000rpm. The pellet was resuspended in 20μL of 1X phosphate buffered saline (PBS) and anti-6x His Alexa Fluor 647 (abcam, cat: ab237337) was added at a dilution of 1:100. Cells were stained in the dark for 1 hour. The treated cells were washed three times in 1X PBS, spinning for 3 minutes at 5000 x g between washes. The final cell pellet was resuspended in 1mL of 1X PBS. FACS was performed on a Beckman Coulter MoFlo Astrios EQ high speed cell sorter using the APC optical configuration (640nm laser, 671/30 bandpass filter). Gates were defined using MB001 pEMH27 cells to gate the "positive/high fluorescent" population and *cmt1*::*zeo* pEMH27 cells were used to define the "negative/low fluorescent" population. After sorting, 1.797 million cells from the positive/high fluorescent and 107.68 thousand cells from the negative/low fluorescent populations were obtained. The populations were diluted 1:10 in 1X PBS and the entire negative/low fluorescent population and approximately one third of the positive/high fluorescent populations were plated to BHI Cam3.5 at 30˚C. Colonies from the positive/high fluorescent population were scraped and stocked after 24 hours of growth at 30˚C. Colonies from the negative/low fluorescent population were scraped and stocked after 64 hours at 30˚C.

## Tn-seq library preparation

Aliquots of the final populations were thawed and the gDNA was extracted using the Wizard Genomic DNA Purification Kit (Promega). Briefly, populations were lysed by: (1) resuspending a cell pellet in 540μL of 50mM EDTA and 60μL of 10mg/mL lysozyme at 37˚C for 30 minutes, (2) addition of 600μL of Nuclei Lysis Buffer (Promega), and (3) incubation at 80˚C for 5 minutes. Cell lysates were treated with 3μL of 4mg/mL RNase solution and incubated at 37˚C for 30 minutes. Proteins were precipitated through the addition of Protein Precipitation Solution (Promega) followed by vigorous vortexing. Samples were incubated on ice for 5 minutes and centrifuged at 16,000 x g for 10 minutes at 4˚C. The supernatant was added to 600μL isopropanol and inverted until a white mass of DNA was visible. Samples were centrifuged for 5 minutes at 16,000 x g and the resulting pellet was washed with 70% ethanol, dried, and rehydrated with DNA Rehydration Solution (Promega) overnight at 4˚C. The genomic DNA (gDNA) was cleaned using the Zymo Genomic DNA Clean and Concentrator kit using the manufacturer specifications for gDNA. The cleaned gDNA was sheared to generate fragments smaller than 600bp and enriched around 300bp. Briefly, 4μg of gDNA was diluted with 10mM

Tris-Cl pH = 8.0 to a final volume of 200μL in low adhesion tubes. The gDNA was sonicated using the following specifications: 20% amplitude, 15 seconds on/15 seconds off duty cycle, 6 minute sonication time (total 12 minutes run time). Proper shearing of the gDNA was confirmed by gel electrophoresis. The sheared DNA was cleaned using AMPure XP beads (Agencourt) following manufacturer specifications. 3'-poly-dC tails were added to the sheared gDNA and filtered through Performa DTR gel filtration cartridges (Edge Biosystems) to remove small molecules. The transposon junctions were amplified using a nested PCR protocol using Easy-A cloning enzyme (Agilent) in which the 3'-poly-dC DNA fragments were amplified, (1) using a forward primer that recognizes the 3'-poly-dC tails (PolyG-1$^{st}$-1) and a reverse primer that recognizes the Tn5 site (Tn5-1$^{st}$-1) and, (2) using a forward primer that recognizes the Tn5 transposon (Tn5-2$^{nd}$-1) and the amplified 5'-poly-gC tail to add a library-specific barcode. The resulting PCR reactions were quantified using Qubit (ThermoFisher Scientific) and equivalent concentrations of the libraries were pooled in a final volume of 30μL. The size range of the pooled PCR products were checked by gel electrophoresis and fragments ranging from 250-500bp were isolated from the gel using a Qiagen Gel Extraction kit. The concentration of the gel-extracted, pooled DNA was measured using Qubit. The size of the DNA fragments and molarity was calculated using the Agilent High Sensitivity D1000 ScreenTape System.

## Tn-seq analysis

The pooled and prepared libraries were run on a MiSeq sequencer (Illumina) and the resulting reads were trimmed using trimmomatic [94] and mapped to the MB001 genome using bowtie 1.0.0 [95]. The further analysis pipeline was performed as described previously [48]. Transposon insertion profiles were visualized using Artemis software [16]. All NGS data from this study are deposited under BioProject PRJNA1113594.

## Flow cytometry

Cells were grown to mid-log (OD600 ~ 0.3–0.8) with aeration in the appropriate media at 30˚C. An equivalent to 1mL of an $OD_{600}$ = 0.3 of cells were gently pelleted (2 minutes at 15,000rpm) and resuspended in 60μL of 1X PBS. anti-6x His Alexa Fluor 647 (Abcam) was added at a dilution of 1:100. Cells were stained in the dark for 45–60 minutes. Samples were washed two times in 100μL of 1X PBS and resuspended in a final volume of 3mL 1X PBS for flow cytometry analysis. Samples were analyzed on a BD LSRII flow cytometry machine using the APC optical configuration (637nm laser, 660/20 bandpass filter).

## Immunoblot analysis

The OD600 of overnight cultures was measured and an equivalent of 1mL of cells at an $OD_{600}$ = 2 of cells was pelleted. The pellets were resuspended in 50μL of lysis buffer (per sample: 0.5μL benzonase/universal protease, 0.5μL 1M MgSO$_4$, 1uL of protease inhibitor cocktail, 10μL of 10mg/mL lysozyme, and 8μL of H$_2$O). Samples were lysed at 37˚C for 1 hour, diluted 1:2 in sample buffer containing β-mercaptoethanol (BME), and boiled for 10 minutes. PorH--His immunoblots were performed on 16.5% tris-tricine gels (BioRad) using anti-His (mouse) primary antibody (GenScript) at a dilution of 1:3,000. RsdA cleavage immunoblot analysis was performed on TGX 10% gels (BioRad) using anti-HA (mouse) primary antibody (Abcam) at a dilution of 1:3,000. In all cases, anti-mouse HRP secondary antibody (Rockland Biosciences) was used at a dilution of 1:3,000.

**β-galactosidase assays**

The $OD_{600}$ of overnight cultures was measured and an equivalent to 1mL at an $OD_{600} = 0.75$ of cells was pelleted. The pellets were resuspended in 1mL of Z-buffer (0.06M $Na_2HPO_4$. $7H_2O$, 0.04M $NaH_2PO_4$. $H_2O$, 0.01M KCl, 0.001M $MgSO_4$, 0.05M BME, pH = 7.0) and incubated at 30°C for 5 minutes. Lysis was completed by adding 5μL of toluene and 20μL of 10mg/mL lysozyme and incubating at 30°C for 15 minutes. A 200μL volume of 4mg/mL O-nitrophenyl-β-D-galactopyranoside (ONPG) solution made up in Z-buffer was added to begin the reaction and the samples were incubated at 30°C. The reaction was quenched with 500μL of 1M $Na_2CO_3$. OD420 and $OD_{420}$ values were read on a Biochrom Ultrospec 2100 pro. Miller units were calculated using the following equation: 1 Miller unit = 1,000 x $(OD_{420}/(time \times volume \times OD_{600}))$. All raw data for β-galactosidase assays are included as S5 Table.

**6TMR-tre fluorescence assay**

Incorporation of 6TMR-tre into the mycomembrane was measured as described previously [68]. Strains were sub-cultured 1:100 in BHIS at 30°C until they reached mid-log (~4 hours). Cells were grown in the presence of a DMSO vehicle control or 0.025mM AU1235, where appropriate. An equivalent to 1mL of cells at an $OD_{600} = 0.3$ was pelleted and resuspended in either BHIS or BHIS supplemented with a 1:100 dilution of 10mM 6TMR-Tre stock solution and incubated at 25°C in the dark for 30 minutes. Samples were washed twice in 1X PBS and resuspended in a final volume of 1mL 1xPBS. Samples were aliquoted into black-walled, clear bottom 96-well plates (Corning). The OD600 of each sample and 6TMR-Tre incorporation was measured at an excitation of 532nm and emission of 580nm on an Tecan Infinite M Plex. Measurements were made in technical duplicate and biological triplicate. All raw data for the 6TMR-tre assay is detailed in S6 Table.

**Fluorescence microscopy**

Overnight cultures of *Cglu* strains were sub-cultured 1:100 in BHI and grown until mid-log (3–4 hours). Cells were concentrated in in 1X PBS and spotted onto 2% M9 agarose pads (. Fluorescent microscopy images were acquired on a Nikon Ti2-E inverted widefield microscope utilizing a motorized stage, a perfect focus system, and a 1.45 NA Plan Apo x100 Ph3 DM objective lens with Cargille Type 37 immersion oil. Images were obtained with Lumencore SpectraX LED Illumination using the mCherry channel with Chroma 49008 filters and an Andor Zyla 4.2 Plus sCMOS camera (65 nm pixel size) with Nikon Elements acquisition software (v5.10). All images were rendered for publication using Fiji and brightness/contrast was normalized between samples to wild-type cells.

## Supporting information

**S1 Fig. Ectopic PorH-His complements *porH* null cells.** Spot titers testing kanamycin sensitivity were prepared by making ten-fold serial dilutions of the indicated cultures and spotting onto media with or without the addition of kanamycin. WT or Δ*porH* cells harbored either an empty vector (EV) or plasmids constitutively expressing PorH or PorH-His. Chloramphenicol was included for plasmid maintenance.
(TIF)

**S2 Fig. Ectopic expression of *cmt1* complements the reduced PorH-His surface exposure of *cmt1::zeo* mutants.** Cells were grown until mid-log and incubated with anti-His Alexa Fluor 647. Stained cells were then washed and analyzed by flow cytometry. Representative data scaled by unit area is shown as a histogram. The strains contain genome-integrated constructs

at the *attB2* site that are either an empty vector or the complement *cmt1* allele induced with 1mM theophylline.
(TIF)

**S3 Fig. Inactivation of the σ$^D$ pathway reduces the level of some proteoforms of PorH-His.** **(A)** The indicated strains without a His-tagged construct were analyzed by immunoblot analysis using a commercial anti-His antibody. Note that native ProtX (Cgp_2785) appears to be detected by the antibody as a non-specific band. **(B)** The indicated strains were analyzed by immunoblot. Red bracket and arrows indicate the PorH-His proteoforms and native ProtX, respectively. Note that ProtX is induced when the σ$^D$ pathway is activated by deletion of *rsdA*. **(C)** Additional two replicates of the same experiment shown in **(B)**.
(TIF)

**S4 Fig. The σ$^D$ pathway does not respond to defects in PG synthesis.** σ$^D$ reporter activity in the indicated strains was measured by β-galactosidase activity. Measurements were made in triplicate and the error bars represent standard deviation.
(TIF)

**S5 Fig. Ectopic overexpression of PorH-His shows protein production.** The indicated strains containing the PorH-His overexpression plasmid (pEMH306) or an empty vector (pEMH309) as well as the σ$^D$ reporter plasmid (pEMH304) were analyzed by immunoblot analysis.
(TIF)

**S6 Fig. MarP and Rip1 are required for σ$^D$ activation.** **(A)** Reporter activity of the listed genetic backgrounds expressing *pks* (black) or with *pks* disrupted (purple) was analyzed by β-galactosidase activity. The average of three replicates is displayed with error bars denoting standard deviation. The displayed numbers indicate the fold change in reporter activity between + *pks* and–*pks* stress condition of the listed genetic background.
(TIF)

**S7 Fig. HA-RsdA complements Δ*rsdA*.** Immunoblot of the indicated genetic backgrounds harboring either an empty vector (EV), a multicopy plasmid expressing untagged RsdA, or a multicopy plasmid expressing HA-RsdA. The anti-His primary antibody detects native ProtX, the production of which is induced upon activation of the σ$^D$ pathway (see S3 Fig).
(TIF)

**S8 Fig. The σ$^D$ pathway is not activated by disruption of LM/LAM biosynthesis.** σ$^D$ activity of controls and LM/LAM biosynthetic mutants was measured by β-galactosidase assay. Data shown are three biological replicates and error bars represent standard deviation.
(TIF)

**S9 Fig. Disruption of trehalose biosynthesis activates the σ$^D$ pathway.** **(A)** Schematic of the three trehalose biosynthetic pathways of *Cglu*. Adapted from [62, 69]. **(B)** β-galactosidase assay of trehalose biosynthetic mutants and controls. σ$^D$ activity of the listed cultures was measured in biological triplicate. Error bars represent standard deviation.
(TIF)

**S10 Fig. AU1235 treatment of Δ*cmpL4* cells strongly reduces TMCM levels.** The listed strains were grown until mid-log with either the vehicle control (DMSO) or 0.025mM AU1235. Equivalent numbers of cells were stained with 100μM 6TMR-Tre for 30 minutes in biological triplicate. The $OD_{600}$ and 6TMR-Tre incorporation were measured and used to calculate relative 6TMR-Tre staining (6TMR-Tre fluorescence emission/$OD_{600}$). Error bars

represent standard deviation.
(TIF)

**S1 Table. Analyzed transposon insertion data for PorH surface exposure Tnseq.**
(XLS)

**S2 Table. Strain List.**
(DOCX)

**S3 Table. Plasmid used in this study.**
(DOCX)

**S4 Table. Oligonucleotides used in this study.**
(DOCX)

**S5 Table. Raw numerical data for β-galactosidase assays.**
(ZIP)

**S6 Table. Raw numerical data for 6TMR-tre assay.**
(ZIP)

## Acknowledgments

The authors are grateful to members of the Bernhardt and Rudner laboratories for advice and helpful discussions, including David Rudner for critical reading of the manuscript and Amelia McKitterick and Anastacia Parks for strains. We would like to thank Paula Montero Llopis and the Microscopy Resources on the North Quad (MicRoN) core facility along with the Immunology Flow Cytometry core at HMS.

## Author Contributions

**Conceptualization:** Elizabeth M. Hart.

**Data curation:** Elizabeth M. Hart.

**Formal analysis:** Elizabeth M. Hart.

**Funding acquisition:** Elizabeth M. Hart, Thomas G. Bernhardt.

**Investigation:** Elizabeth M. Hart.

**Methodology:** Elizabeth M. Hart, Evan Lyerly.

**Project administration:** Thomas G. Bernhardt.

**Resources:** Thomas G. Bernhardt.

**Supervision:** Thomas G. Bernhardt.

**Validation:** Elizabeth M. Hart.

**Visualization:** Elizabeth M. Hart, Thomas G. Bernhardt.

**Writing – original draft:** Elizabeth M. Hart.

**Writing – review & editing:** Elizabeth M. Hart, Thomas G. Bernhardt.

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
