## [Decision Letter · Decision Letter 0]

29 Jan 2024

Dear Dr BERNHARDT,

Thank you very much for submitting your Research Article entitled 'The conserved σ^D^ envelope stress response monitors multiple aspects of envelope integrity in corynebacteria' to PLOS Genetics.

The manuscript was fully evaluated at the editorial level and by independent peer reviewers. The reviewers appreciated the attention to an important topic but identified some concerns that we ask you address in a revised manuscript.

We therefore ask you to modify the manuscript according to the review recommendations. Your revisions should address the specific points made by each reviewer.

Yours sincerely,

Ankur B. Dalia

Academic Editor

PLOS Genetics

Sean Crosson

Section Editor

PLOS Genetics

Editor's Comments:

As you can see below, all three reviewers were enthusiastic about the initial submission. While most of the comment can be addressed by changes to the text and figures, there are a few points that may require additional experiments. In particular, complementation of key strains is a point that was raised by two reviewers. If these points can be thoroughly addressed, I would welcome a revised draft of the manuscript.

Reviewer's Responses to Questions

**Comments to the Authors:**

Reviewer #1: This is a well-written study that will interest the cell envelope field. I especially appreciate the effort that went in to trying to narrow down the specific envelope defects that induce the sigma D stress response.

Comments:

1) Are beta gal assays complementable? This is a relatively minor concern as there are lots of different mutants and treatments to test each hypothesis.

2) First paragraph of Discussion is helpful, especially the last sentence, and I wonder if the authors could have a similar sentence somewhere at the end of the Introduction to help orient the reader and set expectations.

3) Many of the deletions are in cell envelope genes that seem like they should be essential, at least by analogy to other organisms. Can the authors address this point somewhere?

4) Fig. 4 legend missing panels (E) and (F)

Reviewer #2: Hart et al. identified proteins responsible for mycolate outer membrane protein (MOMP) insertion into the outer mycomembrane in Corynebacterium glutamicum. These MOMPs may function like porins from Gram-negative bacteria, but their role in corynebacteria and mycobacteria is largely unknown. The authors identified proteins important for surface exposure of a MOMP named PorH. Using a FACS screen on a transposon-mutagenized library for mutants that alter surface exposure of PorH, the authors identified the alternative sigma factor sD, two proteases that liberate sD from its anti-sigma factor RsdA, and the Cmt1 enzyme that transfers mycolic acids to MOMPs. The sD regulon has been shown to be involved in an envelope stress response, and includes many enzymes important for synthesis of the envelope layers. Using a LacZ reporter, the authors narrowed down genes and antibiotics that contribute to sD activation. They rule out peptidoglycan inhibition as an activation signal, and clearly define which steps of both arabinogalactan and mycolic acid synthesis and transport are recognized by the sD pathway. This study will be of interest to those studying the corynebacterial or mycobacterial envelope organization and assembly.

The major conclusions of this paper are:

1. Loss of the alternative sigma factor sD (encoded by sigD), or either of the proteases rip1 or marP, diminishes surface exposure of the MOMP PorH.

2. Activation of the pathway is dependent on both proteases rip1 and marP. Further genetic and antibiotic treatments reveal that loss of mycolic acid synthesis or arabinogalactan (AG) synthesis, but not peptidoglycan (PG) synthesis, are activating signals.

3. MarP is the previously unidentified site-1 protease required for degradation of anti-sigma factor RsdA, and sD activation requires both MarP and Rip1 (the known site-2 protease)

4. A series of genetic knockouts in the AG synthesis pathway reveals that sD specifically responds to loss of the core AG chain, but not loss of AG branching (Fig 4).

5. A series of genetic knockouts in the mycolic acid synthesis pathway reveals sD specifically responds to loss of mycolic acids on the plasma membrane, but not loss of mycolic acid transport to the outer mycomembrane (Fig 5).

This paper has several major strengths that contribute to our understanding of MOMP export and the envelope stress response in the order Mycobacteriales, which includes several major human pathogens. The initial transposon screen was effective in identifying the sD pathway, including a protease previously unknown to cleave the sD’s anti-sigma factor RsdA. Follow-up experiments clearly and concisely define the steps in envelope synthesis or transport that are detected by sD. The authors utilized a broad range of both genetic knockouts and drug treatments to narrow down the specific defects in the envelope that activate sD. Furthermore, the authors made very clear pathway schematics that made it easy to follow the synthesis steps being tested.

I just have a few suggestions for the authors:

1 - Overall, while the data are very convincing on their own, most figures would benefit from ANOVA statistical analysis, as in Figure 1A. For example, there appears to be some intermediate activation of the lacZ reporter in Figure 4F for loss of aftB.

2 - Page 8:

The logic on Page 8 is confusing around the unexpected decrease in signal upon expression of the second copy of cmt1. I suggest rewriting this section to clarify the logic of the experiments and the conclusion.

In addition to this, the statements around the PorH proteoforms in Figure S3 are not convincing. Claiming a difference in the upper two proteoforms for ∆sigD in S3B is difficult without loading controls, replicates, and quantification. I suggest either softening this claim or providing more evidence.

Overall, the conclusions from Figures 2D and S3 feel a little disjointed. The rest of the paper was clearly written and the experiments were explained nicely

3 - The figure call to Figure S4 mentions inductions at different stages of growth, but this is not shown.

Page 18:

missing a period at sentence ending “…defects in AG biosynthesis that promote sD activation”

Reviewer #3: In this manuscript, the authors aimed to unravel the transportation mechanisms of Mycolate Outer Membrane Proteins (MOMP) in Corynebacteria. They employed the staining of the outer region of MOMP using a His-tag as a readout for MOMP transport. Although the authors used this staining method to identify genes important for MOMP transporter, interestingly, their transposon-sequence analysis led them to identify genes related to sigmaD activation and sensing of cell envelope defects. While it was observed that the protein regulated by sigmaD appeared to play a role in MOMP transport, the authors redirected their focus towards understanding the mechanisms behind sigmaD activation, which had not been previously investigated in Mycobacteriales. The authors successfully uncovered that MarP functions as the site-1 protease, and Rip1 serves as the site-2 protease responsible for cleaving RsdA to activate sigmaD. This cascade is initiated by sensing the precursor of arabinogalactan and/or the absence of TMM in the plasma membrane. While these findings shed light on important sensing mechanisms related to cell envelope defects, there are some results that require further experimental validation.

Major Comments:

1. The authors suggest that sigmaD responds to “loss of the arabinan chain” and, in the Discussion, they further speculate that “RsdA might use its long, disordered periplasmic domain of approximately 160 amino acids to detect AG defects”. However, it seems equally likely that RsdA senses the accumulation of lipid-linked galactan core in the inner membrane. The authors should provide additional experimental evidence to support this speculation or tone down with their claim that RsdA is sensing the loss of arabinan chain.

2. Without providing the complemented strain for the emb::kan and aftA::kan strains, it remains possible that sigmaD activation is due to a polar effect of emb or aftA gene deletion. Since RsdA may be sensing inner membrane defect (such as the absence of TMM as the authors suggest), it is even possible that RsdA is sensing the absence of Emb or AftA proteins rather than the defects in arabinans. These mutants should be complemented minimally by wildtype genes, and ideally also by catalytically inactive versions of the corresponding genes.

3. Similarly, ethambutol may have secondary side effects on the inner membrane homeostasis. If emb::kan is complemented by ethambutol-resistant version of emb, and if ethambutol does not induce sigmaD activation in the complemented strain, that would strengthen the authors’ claims. Minimally, the authors should discuss these alternative possibilities.

4. Provide the gene ID of ProtX. Provide how the mutant was generated, including the plasmids and oligos used to generate it.

5. The authors stated “In the absence of Cmt1, PorH-His cannot be O-mycoloylated, leading to a dramatic decrease in PorH levels in the MM (6–8, 12). As anticipated, cmt1::zeo cells expressing PorH-His showed no surface fluorescence (Figure 1A)”. This statement is misleading as the authors later show that PorH is not even expressed at all in a cell lysate of the cmt1 mutant (Fig. S3). If the authors truly intend to make the claim that PorH level is reduced in the MM, but accumulating elsewhere, there should be supporting evidence for the claim.

Minor Comments:

1. In Corynebacteria, TMM and TDM should be described as trehalose monocorynomycolate (TMCM) and TDCM.

2. In Fig. 1, the Cmt1 mutant should be complemented to confirm that the observed phenotype is indeed due to the deletion of cmt1.

3. On page 4, Line 19, the argument “... the biogenesis of which is targeted by front-line anti-mycobacterial therapies. "needs references.

4. On page 6, Line 13, Cmt1 was not annotated in the cited papers. Indicate the gene number of Cmt1 to help orient readers.

5. On page 10, line 2, confirm porH overexpression through immunoblotting. Otherwise, the argument “No induction was observed when the MOMP was overproduced” cannot be supported.

6. In Figure 2B, modify the graph label to make it clear that cgp_0697 is rsdA.

7. In Figure 3D, indicating the expected molecular weight of the full-length and cleaved protein would be helpful for readers.

8. Regarding Figure S9, it appears that AU1235 increased the relative fluorescence of 6TMR-Tre. Please clarify why this is happening.

9. Indicate the panel when referring to Fig. S3 in the text.

**Have all data underlying the figures and results presented in the manuscript been provided?**

Reviewer #1: Yes

Reviewer #2: Yes

Reviewer #3: **No: **Lacking information on ProtX. See the critique above.

PLOS authors have the option to publish the peer

---

## [Editor Report · Decision Letter 1]

15 May 2024

Dear Dr Bernhardt,

We are pleased to inform you that your manuscript entitled "The conserved σ^D^ envelope stress response monitors multiple aspects of envelope integrity in corynebacteria" has been editorially accepted for publication in PLOS Genetics. Congratulations!

Yours sincerely,

Ankur B. Dalia

Academic Editor

PLOS Genetics

Sean Crosson

Section Editor

PLOS Genetics

**Data Deposition**

http://datadryad.org/submit?journalID=pgenetics&manu=PGENETICS-D-24-00028R1

**Press Queries**

---

## [Editor Report · Acceptance letter]

28 May 2024

PGENETICS-D-24-00028R1 

The conserved σ^D^ envelope stress response monitors multiple aspects of envelope integrity in corynebacteria 

Dear Dr Bernhardt, 

We are pleased to inform you that your manuscript entitled "The conserved σ^D^ envelope stress response monitors multiple aspects of envelope integrity in corynebacteria" has been formally accepted for publication in PLOS Genetics! Your manuscript is now with our production department and you will be notified of the publication date in due course.

With kind regards,

Olena Szabo

PLOS Genetics

On behalf of:
